# The Impact of Improved Spatial and Temporal Resolution of Reanalysis Data on Lagrangian Studies of the Tropical Tropopause Layer

Stephen Bourguet[1] and Marianna Linz[1, 2]

[1]Harvard University Department of Earth and Planetary Sciences
[2]Harvard University School of Engineering and Applied Sciences

**Correspondence:** Stephen Bourguet (stephen_bourguet@g.harvard.edu)

**Abstract.** Lagrangian trajectories are frequently used to trace air parcels from the troposphere to the stratosphere through the tropical tropopause layer (TTL), and the coldest temperatures of these trajectories have been used to reconstruct water vapor variability in the lower stratosphere, where water vapor's radiative impact on Earth's surface is strongest. As such, the ability of these trajectories to accurately capture temperatures encountered by parcels in the TTL is crucial to water vapor reconstructions and calculations of water vapor's radiative forcing. A potential source of error for trajectory calculations is the resolution of the input data. Here, we explore how improving the spatial and temporal resolution of model input data impacts the temperatures measured by Lagrangian trajectories that cross the TTL during boreal winter using ERA5 reanalysis data. We do so by comparing the temperature distribution of trajectories computed with data downsampled in either space or time to those computed with ERA5's maximum resolution. We find that improvements in temporal resolution from 6- to 3- and 1-h lower the cold point temperature distribution, with the mean cold point temperature decreasing from 185.9 to 185.0 and 184.5 K for reverse trajectories initialized at the end of February for each year from 2010 to 2019, while improvements to vertical resolution from that of MERRA2 data (the GEOS5 model grid) to full ERA5 resolution also lower the distribution but are of secondary importance, and improvements in horizontal resolution from $1° \times 1°$ to $0.5° \times 0.5°$ or $0.25° \times 0.25°$ have negligible impacts to trajectory cold points. We suggest that this is caused by excess vertical dispersion near the tropopause when temporal resolution is degraded, which allows trajectories to cross the TTL without passing through the coldest regions, and by undersampling of the four–dimensional temperature field when either temporal or vertical resolution is reduced.

## 1 Introduction

The composition of air entering the middle atmosphere through the tropical tropopause layer (TTL) is an important control on the composition of air throughout the stratosphere. This idea was proposed by Alan Brewer in 1949 to explain his observations of a dry mid-latitude stratosphere: the coldest region that mid-latitude air may have encountered is near the tropical tropopause (Fig. 1), so the air must pass through that layer to achieve its level of dehydration (Brewer, 1949). Additional tracer observations have confirmed the existence of this overturning circulation, now known as the Brewer–Dobson circulation (BDC) (Dobson, 1956; Newell, 1963). Subsequent work has confirmed the importance of the TTL in setting the humidity of the stratosphere

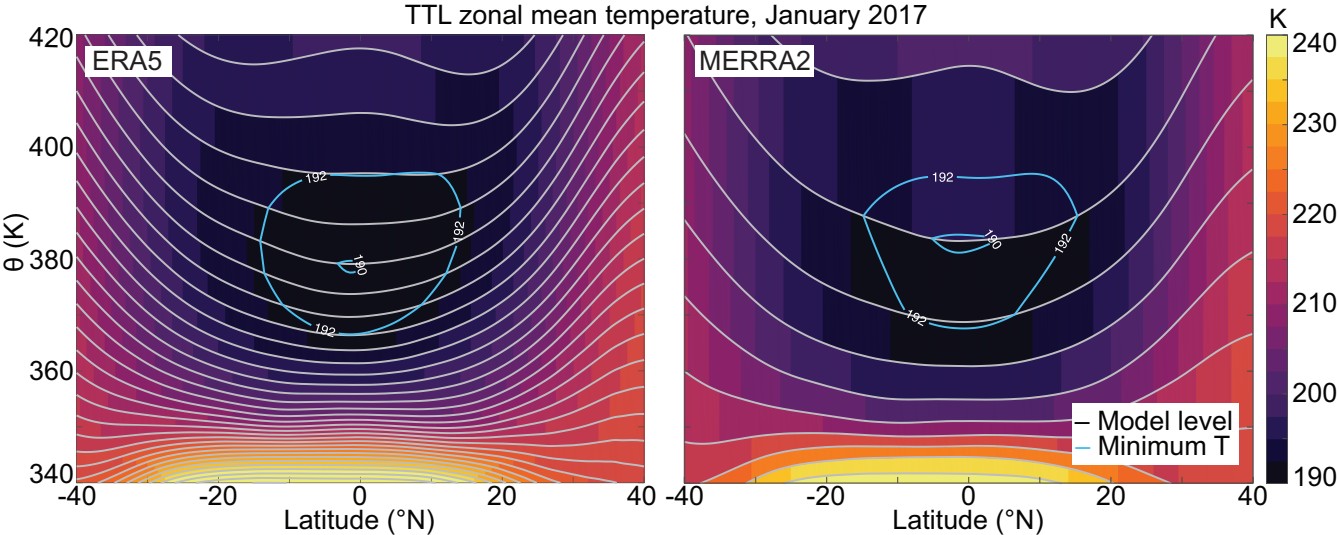

**Figure 1.** The January 2017 zonal mean tropical tropopause temperature for ERA5 and MERRA2 reanalysis data. Note the increased number of model layers near the coldest temperatures for the ERA5 data. A plot of the reduced resolution ERA5 data used in this paper is shown in Fig. S1.

by connecting seasonal and interannual temperature fluctuations in the TTL to water vapor variability at heights throughout
the lower and middle stratosphere with a time lag corresponding to the anomaly's transit time via the background circulation
(the "water vapor tape recorder," Mote et al. (1996); Randel and Park (2019)). This correlation between local water vapor and
the tropopause concentration at an earlier time is strong up to about 30 km in the tropics, at which point mixing and input of
water vapor from the oxidation of methane dilutes the anomaly coming from the cold point. Quasi-horizontal transport along
isentropes also connects TTL temperatures to extratropical water vapor above 13 km (Randel and Park (2019)). Although the
lower stratosphere is very dry, water vapor's radiative forcing is strongest in this layer, so small changes in its humidity can
have a large impact on the climate (Solomon et al., 2010).

   This two-dimensional description accurately describes the meridional movement of air in the stratosphere, but it does not
capture zonal variability in the circulation or in troposphere–to–stratosphere transport (TST). A seasonally varying zonal struc-
ture for TST was proposed by Newell and Gould-Stewart (1981) based on global 100 mb temperatures to maintain the low
humidity of the stratosphere (the "stratospheric fountain"). This framework proposes that air enters the stratosphere over the
Tropical West Pacific during boreal winter and over the Bay of Bengal and India during the boreal summer, with the majority
of the flow into the stratosphere occuring during boreal winter. Subsequent work by Holton and Gettelman (2001) countered
this hypothesis by contrasting the vertical and zonal velocities in the TTL: the vertical velocity is orders of magnitude smaller
than the zonal velocity, so air can circulate within the layer and encounter the "cold trap" far from where it ultimately enters
the stratosphere. The stratospheric fountain and cold trap hypotheses both propose that zonal variability in the structure of
TTL is important for understanding the composition of the lower stratosphere. As mentioned above, the water vapor mixing

ratio at entry into the stratosphere is determined by the extent of dehydration within the TTL, and concentrations of other key species with regional sources depend on the tropospheric origin of stratospheric air masses and their residence time within the TTL (Fueglistaler et al., 2009; Randel et al., 2007). Stratospheric temperatures and circulation are also strongly impacted by the local radiative forcing of water vapor and ozone (Maycock et al., 2011, 2013; Ming et al., 2017). Therefore, an accurate representation of the TTL's three-dimensional structure is necessary for understanding the composition and circulation of the stratosphere.

Lagrangian trajectory models have also been used to show that the coldest temperature of trajectories that transit the TTL are predominantly encountered over the Tropical West Pacific, and that entry into the stratosphere occurs at least 20 days later and thousands of kilometers away (Fueglistaler et al., 2004, 2005; Liu et al., 2010; Schoeberl and Dessler, 2011; Schoeberl et al., 2012, 2013). These models also show that the region controlling trajectory dehydration shifts to the Indian subcontinent and southeast Asia during boreal summer, consistent with the observations used to hypothesize the stratospheric fountain. Bowman et al. (2013) found that reanalysis products used for Lagrangian trajectory modeling available at the time of publication were deficient in temporal resolution relative to their spatial resolution. Reanalyses were available on 0.5° horizontal grids at 3- or 6-h temporal resolution, which caused Lagrangian trajectory models to undersample the temporal variance of meteorological fields and not take advantage of the improved spatial resolution of the input data. Pisso et al. (2010) found that trajectories run on a 1° horizontal grid improved when the temporal resolution was increased from 6- to 3- and 1-h, with the improvement from 6- to 3-h being much greater than that from 3- to 1-h. Wang et al. (2015) found that running a Lagrangian trajectory model with temperature data enhanced vertically to match GPS observations or with a correction for finer scale waves did not have a significant impact on the water vapor predicted from dehydration at the Lagrangian dry point. Therefore, improving the temporal resolution of input data for Lagrangian trajectory models may have the greatest potential to improve model performance, though improved vertical resolution may also have an effect.

Another important consideration for Lagrangian trajectory models is whether the input vertical velocities are diabatic or kinematic. Previous work has shown that kinematic trajectories are more dispersive than diabatic trajectories (Schoeberl (2004); Wohltmann and Rex (2008); Schoeberl and Dessler (2011)). Liu et al. (2010) found that this excess dispersion impacts a model's prediction of water vapor in the stratosphere based on the saturation mixing ratio of water vapor at the trajectories' Lagrangian dry point, but that this effect decreased with increased temporal resolution and with an updated reanalysis product (ERA-interim vs. ERA-40). Li et al. (2020) showed that calculations of transport to the tropopause layer by tropical cyclones using both diabatic and kinematic vertical velocities are improved when using ERA5 data in place of ERA-interim due to the improved spatial and temporal resolution of the updated reanalysis. The kinematic ERA5 trajectories in Li et al. (2020) capture the same cold temperatures associated with convection as the diabatic trajectories, and they represent the cyclonic air motion better than the diabatic trajectories, while the ERA-interim kinematic trajectories were not able to do so. It is therefore possible for kinematic vertical velocities provided at a high enough resolution to minimize the dispersive errors reported with older data products. This would be ideal for future studies because kinematic vertical velocities are more widely available than diabatic vertical velocities as outputs from reanalyses and models, and diabatic vertical velocities are often provided as daily or monthly means.

The stratospheric water vapor values calculated from the temperature history of the aforementioned Lagrangian trajectories studies are in good agreement with satellite observations, but it is not inconceivable that these values are right due to compensating errors. The undersampling of the temperature field due to insufficient spatial or temporal resolution is bound to produce warm (moist) biases in the cold point temperature distribution, while an underestimation of the fraction of air originating in the troposphere due to unrealistic trajectory paths would decrease the mass of air recently undergoing dehydration and produce an additional moist bias. Therefore, improving the models' representation of how air parcels transit the TTL by increasing the temporal resolution of input data could eliminate this moist bias and decrease the water vapor concentration calculated from Lagrangian trajectory models. This would imply that an additional source of water vapor is needed to match the observed values. Previous work has suggested that ice lofting could potentially inject a significant amount of water into the lower stratosphere (Keith, 2000; Schoeberl and Dessler, 2011), as evidenced by the high observed ice water path over the central Pacific in the TTL during the 2015–2016 El Niño (Avery et al., 2017). An additional source of water vapor could also come from the ascending air's relative humidity exceeding 100% with respect to ice due to condensation being limited by insufficient ice nucleating particles or by cloud microphysical processes (Schoeberl et al., 2014, 2016; Ueyama et al., 2015). These hypotheses for the source of additional water vapor have flaws though: isotopic constraints have been used to suggest that ice lofting brings water vapor to the upper troposphere but not across the cold point tropopause, which means that ice lofting cannot inject significant water vapor to the lower stratosphere (Dessler et al., 2007). A similar conclusion was found using dynamic constraints by Bolot and Fueglistaler (2021), who showed that convective ice lofting supplies water up to the cold point tropopause but the transition to the slow ascent transport regime above that layer prevents further upward motion of significant amounts of ice. Meanwhile, the excess humidity of air entering the stratosphere may be limited by gravity waves and aging cirrus anvils, which improve the cloud dehydration efficiency by increasing the ice particle count (Schoeberl et al., 2015; Ueyama et al., 2020). Therefore, the extent to which previous Lagrangian trajectory studies of water vapor in the lower stratosphere have been biased by insufficient resolution and an inability to represent the physical processes that control lower stratospheric water vapor is an important open question. This also relevant to a recent water vapor reconstruction from a Lagrangian trajectory model that shows a moistening of the stratosphere since the year 2000 (Konopka et al., 2022).

With the release of ECMWF's updated ERA5 (Hersbach et al., 2020), we can now run Lagrangian trajectory models with 1-h resolution and $0.25° \times 0.25°$ horizontal resolution on a 137-level vertical grid to analyze the improvements of kinematic trajectory models with enhanced temporal and spatial resolution. Below, we described our model setup (Section 2), show the results of our model integrations with a focus on dispersion and its impact on analysis of TTL temperatures and water vapor (Section 3), and discuss the implications of these results for future Lagrangian trajectory studies (Section 4). While improving this Lagrangian trajectory method does not directly enhance our understanding of the processes that set the the humidity of air entering the stratosphere, it does give a more accurate estimation of the extent to which processes beyond a simple cold point dehydration mechanism must be considered in calculating this value.

## 2 Methods

 ### 2.1 Model and data

We use ECMWF's (European Center for Medium-range Weather Forecasts) Lagrangian analysis tool LAGRANTO version 2 (Sprenger and Wernli, 2015), which has been updated from its precursor with more flexible ways to initialize and select trajectories, with ERA5 (ECMWF's latest reanalysis) input data (Hersbach et al., 2020). This data is available on a $0.25° \times 0.25°$ horizontal grid on 137 native model levels in 1-h timesteps. When converted to pressure levels using a fixed surface pressure of 1013.25 hPa, ERA5 has 20 levels between 200 and 70 hPa, which is an improvement from the 6 levels that ERA-interim and other reanalyses (MERRA-2 and JRA-55) have in this range (Tegtmeier et al., 2020). LAGRANTO computes parcel trajectories by integrating the velocity equation forward or backward through time using the three-dimensional kinematic wind field (vertical velocity is in units of Pa s$^{-1}$). The wind vector is averaged between timesteps before integrating the trajectory forward, and spatial interpolation is done using bilinear interpolation in the horizontal and linear interpolation in the vertical. The trajectories are calculated on pressure levels, and the temperature and potential temperature are recorded along the trajectories' paths. As mentioned above, trajectories run with kinematic vertical velocities are overly dispersive relative to those run with diabatic heating rates, but this effect decreases with increased resolution and an improved data product (Liu et al., 2010; Li et al., 2020). We acknowledge that this may still introduce some error in these calculations, but we show that it is minimal due to the high resolution of the data (see Section 3.1.2).

By default, LAGRANTO integrates trajectories 12 times per input data timestep (e.g. every 5 minutes for 1-h data or 30 minutes for 6-h data). To test the sensitivity of trajectories to the length of the integration timestep, we ran a set of integrations with 1-h data and a 30-minute timestep and a set with 6-h data and a 5-minute timestep. The results are nearly identical to the integrations run with LAGRANTO's default timesteps (Fig. S2), so we ran trajectories with the default setting, and we obtained outputs once per hour regardless of the input data frequency.

 ### 2.2 Lagrangian cold point analysis

We test how LAGRANTO observes the TTL cold point using a range of spatial and temporal resolutions as described in Table 1. The figures presented in Section 3 contain data from DJF 2017, which is close to the average of all years; throughout the text we note when we are quoting either 2017 data or averages from $2010 - 2019$. All data was obtained at 1-h, $0.25° \times 0.25°$ or $0.5° \times 0.5°$, and 137-level resolution. The temporal resolution was decreased by subsampling the instantaneous data every third or sixth hour (not averaging over these time periods), and the vertical and horizontal resolutions were reduced using Climate Data Operators' remapping tools, remapeta and remapcon (Schulzweida, 2021). We chose to reduce the ERA5 data to the vertical levels closest to those of MERRA2 rather than use MERRA2 data for this comparison to prevent differences in the meteorological fields of the datasets from biasing the results.

For each configuration listed in Table 1, we initialize 5 sets of trajectories between 5° S and 5° N at all longitudes with 0.5° spacing for a total of over 75,000 trajectories per experiment. The trajectories are run on pressure levels, so the trajectories' starting heights are interpolated to the pressure levels corresponding to the 400 K isentrope in order to track how air parcels

| Timeframe | Temporal resolution | Horizontal resolution | Vertical resolution |
|-----------|---------------------|----------------------|---------------------|
| DJF 2017 | 1 hour | $0.5° \times 0.5°$ | 137 levels |
| DJF 2017 | 3 hour | $0.5° \times 0.5°$ | 137 levels |
| DJF 2017 | 6 hour | $0.5° \times 0.5°$ | 137 levels |
| DJF 2017 | 1 hour | $0.25° \times 0.25°$ | 137 levels |
| DJF 2017 | 1 hour | $1.0° \times 1.0°$ | 137 levels |
| DJF 2017 | 1 hour | $0.5° \times 0.5°$ | 72 levels |
| DJF 2010 – 2019 | 1 hour | $0.5° \times 0.5°$ | 137 levels |
| DJF 2010 – 2019 | 3 hour | $0.5° \times 0.5°$ | 137 levels |
| DJF 2010 – 2019 | 6 hour | $0.5° \times 0.5°$ | 137 levels |

**Table 1.** Summary of the LAGRANTO runs used to test the sensitivity of trajectory cold points to input data resolution. All runs are with ERA5 data (Hersbach et al., 2020), which was downloaded at 1-h, $0.25° \times 0.25°$ or $0.5° \times 0.5°$, and 137-level resolution and subsampled or remapped using Climate Data Operators tools.

reach this level. For the 72-level vertical resolution trajectories, we initialize trajectories on the 75 hPa pressure level due to issues with LAGRANTO's interpolation to the 400 K isentrope when using the 72-level data. A set of trajectories is initialized at the final timestep of each day from February 24 to 28 and was integrated backwards to December 1 of the previous year
(integrations last between 86 and 90 days depending on start day). The trajectory cold point is taken at the coldest temperature recorded during the integration. We also ran one set of integrations between 15° S and 15° N and determined that expanding the trajectories beyond the deep tropics was not necessary for our analysis (Fig. S3). We found significant differences between the results from the two vertical resolutions, so we used the full vertical grid for the horizontal and temporal resolution comparisons (see Section 3.2 for vertical resolution comparison). In contrast, we found that our analysis did not significantly change when
the horizontal resolution was increased from $1.0° \times 1.0°$ to $0.5° \times 0.5°$, and it changed even less when increased to $0.25° \times 0.25°$, consistent with Bowman et al. (2013). Therefore, we performed the vertical and temporal resolution comparisons with $0.5° \times 0.5°$ horizontal resolution (see Section 3.3 for horizontal resolution comparison).

We determined that the integration length was sufficient for this study based on the convergence of the cold point temperature and height distributions by the end of the runs (Fig. S4). The fraction of trajectories traced to the troposphere also asymptotes
for the 6-h trajectories by day 90, but the fraction for the 1-h trajectories does not reach an asymptote within the integration. Therefore, we cannot comment here on the fraction of trajectories at 400 K that ultimately ascend from the troposphere, but we are still confident in these trajectories' representation of the cold point based on Fig. S4. We do introduce a bias in the observed cold points towards the TTL conditions in late January and early February by initializing at the end of February, although trajectories still need to be run through December to trace them to the troposphere. As we will discuss in Section 3,
this bias differs for each set of trajectories and needs to be considered when comparing trajectory cold points.

We chose to analyze boreal winter because the Brewer-Dobson circulation and the correlation between TTL temperatures and lower stratospheric water vapor are both strongest during this season (Rosenlof, 1995). We computed 1-, 3-, and 6-h

trajectories for DJF 2010 to DJF 2019 (with the year corresponding to the JF year), and found that our temporal resolution results were robust to the range of natural variability exhibited over the decade. We then performed our spatial resolution analysis with DJF 2017 data because of that year's neutral ENSO state and because the cold point statistics from that year are close to the mean statistics from the decade. The temporal resolution comparisons for the other years can be found in the Figs. S6-8.

Following the method of Fueglistaler et al. (2005), we use trajectories that are traced below 340 K for our analysis of the Lagrangian cold point. This set of trajectories consists of 32–75% of the total initialized trajectories depending on the year and the resolution of the input data; interannual variability accounts for the majority of this variance, while input data resolution only drives a few percent change. For the DJF 2017 runs listed in Table 1, 53% of the 1-h trajectories run at each horizontal resolution, 58% of the 3-h trajectories, 61% of the 6-h trajectories, and 46% of the 72-level vertical resolution trajectories were traced below 340 K. (The 72-level trajectories have a smaller fraction due to their 75 hPa starting height.) Our results do not change if the set of trajectories is expanded to include those that go below 360 K but not 340 K. For each trajectory in this subset, we find the cold point's temperature, pressure, and longitude, and we calculate the trajectory's transit times from 400 K to the cold point and to 340 K. We then create distributions of their values using probability density functions. Finally, we estimate the water vapor concentration at entry to the stratosphere using the saturation water vapor at the cold point taken from the Clausius–Clapeyron relationship assuming 100% relative humidity with respect to ice.

### 2.3 Dispersion and the tropopause transport barrier

To explore the well documented vertical dispersion of kinematic trajectories, we run a set of 20-day reverse trajectories on a global $0.5° \times 0.5°$ horizontal grid initialized on pressure levels corresponding to isentropes every 10 K between 310 K and 420 K starting on the final day of January 2017 with 1-, 3-, and 6-h temporal resolution. We then calculate the zonal mean and zonal variance (i.e. the variance across longitudes at a given latitude and height) of the trajectories' final displacement on each starting isentrope. We assume that excess dispersion should not be biased in either direction, so we hypothesize that the zonal mean of the displacement should be similar across temporal resolutions while the zonal variance should have a direct relationship with dispersion. This difference in variance should be largest near a transport barrier, where a proper representation of the flow should yield a low variance in displacement (the range of trajectory motion is limited by the barrier), while a poor representation of the barrier due to dispersion would allow trajectories to move to a greater range of displacements. Therefore, the variance of displacement should be low in the lower stratosphere, where the gradual ascent of air through the TTL and gradual descent of air at higher latitudes should limit the range of trajectory displacements. Increases in variance in this region would indicate that excess dispersion interferes with the accurate representation of TST and/or the Brewer–Dobson circulation.

To look more closely at trajectory dispersion in the deep tropics, we follow the height of reverse trajectories initialized above and below the TTL transport barrier (interpolated onto 400 K and 340 K isentropes) over the course of 90 day integrations. With increased dispersion, we expect both a larger fraction of trajectories to more quickly cross the TTL and a larger fraction of trajectories to be traced upwards into the stratosphere.

# 3 Results

## 3.1 Temporal Resolution

### 3.1.1 Cold Point Temperature

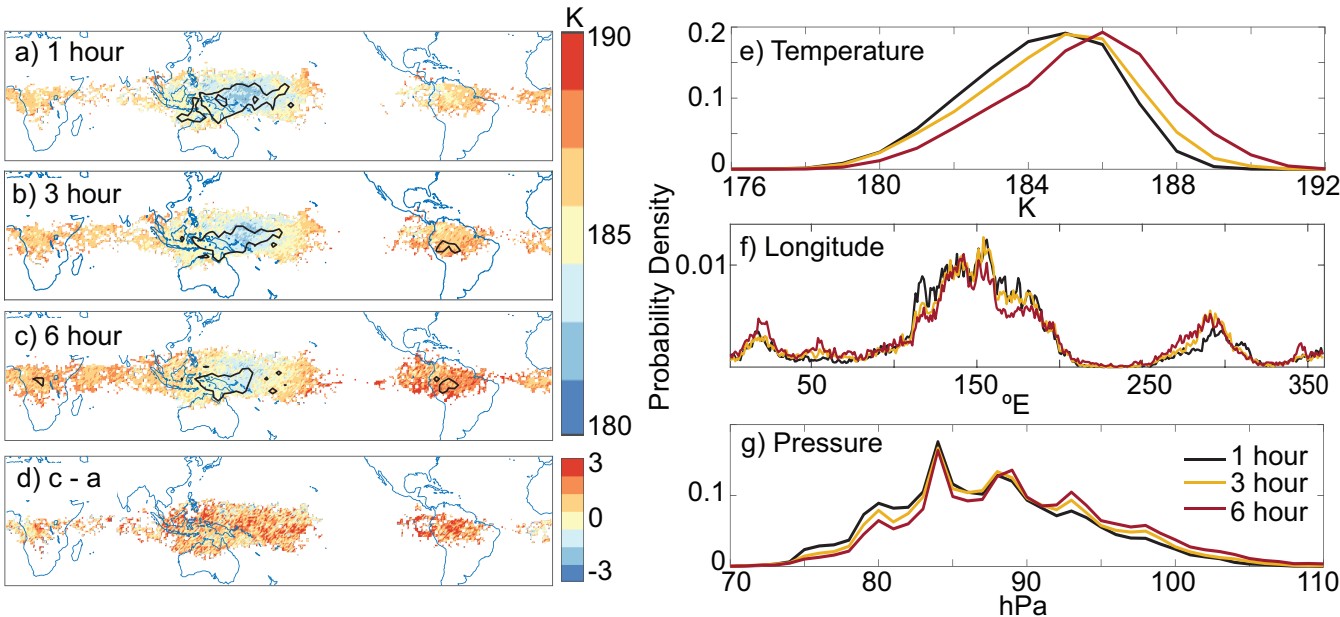

**Figure 2.** (a – d) Spatial distribution of the DJF 2017 trajectories' cold point temperature for 1-, 3-, and 6-h, and the difference in temperature between the 6- and 1-h trajectories. Black contours in a), b) and c) denote regions where the fraction of trajectories crossing grid cells is greater 0.03%. (e – g) Probability density functions of the cold point temperature, longitude, and pressure for 1-, 3-, and 6-h trajectories.

Figure 2 displays the location and temperature of the cold point of DJF 2017 trajectories run with 1-, 3-, and 6-h input data
(panels a–c), the temperature difference (6-h cold points − 1-h cold points) for locations where both the 1- and 6-h trajectories experience a cold point (panel d), and probability density functions (PDFs) of the cold point temperature, longitude, and pressure (panels e–g). For the distributions in Fig. 2a–c, the cold point locations are binned into 1° boxes and the average of all cold point temperatures within each box is taken; the difference of these averages for boxes where both 6-h and 1-h trajectories register at least 1 cold point (i.e. the cold points are colocated) is shown in Fig. 2d. The contours in Fig. 2a–c
denote the regions where the 1° boxes contain greater than 0.03% of the cold points observed by trajectories that are traced to the troposphere. (This is five times what the probability would be if the cold points were uniformly spaced across the tropics.) We chose to display the comparison of 1- and 6-h trajectories in Fig. 2d because of the frequent use of 6-h resolution in previous studies (Fueglistaler et al., 2005; Konopka et al., 2022), but the same conclusions come from analysis done with comparisons involving 3-h data. Fig. 2a–c do not consider cold point height, although it generally decreases with increasing

temperature and its changes follow the same spatial pattern as temperature (see Fig. S9 for corresponding maps of cold point pressure). The mean cold point temperatures for 1-, 3-, and 6-h resolution trajectories, as is shown in Fig. 2e, are 184.8, 185.1, 186.0 K, respectively. (The mean cold point temperatures across DJF 2010 – 2019 trajectories are 184.5, 185.0, and 185.9 K. Corresponding PDFs for these years are shown in Figs. S6–8) There is also a shift in the mean cold point pressure from 87.4 hPa for the DJF 2017 1-h trajectories to 88.3 and 89.4 hPa for the 3- and 6-h trajectories, and although the majority of trajectories that experience their cold point in the West Pacific cold trap (120°E to 200°E) for all runs, this fraction decreases from 63.5% for the 1-h trajectories to 60.5% and 53.9% for 3- and 6-h trajectories.

A major source of these errors is the undersampling of the wind field by lower resolution trajectories. As we will discuss in Section 3.1.2, the decreased temporal sampling of the wind field results in excess dispersion in the lower stratosphere and TTL, which causes trajectories to undersample spatial temperature variability by crossing the TTL before reaching regions with lower temperatures and/or by skipping over the coldest point along a trajectory's path. We can approximate the resulting warm bias by reweighting the lower resolution trajectories' cold point temperatures with the spatial distribution of the 1-h trajectories. This removes "edge" cold points, which are located outside of the horizontal range of the 1-h trajectories and therefore get a weight of 0 in this redistribution. Previous work with 6-h temporal resolution noted the importance of these edge cold points in determining the mean cold point (Schoeberl et al., 2013). The mean temperature of these edge cold points is 187.2 and 187.9 K for the 3- and 6-h trajectories, respectively, while the temperatures in the colocated regions are 185.0 and 185.7 K. The edge points comprise about 8% and 11% of the trajectories analyzed for the 3- and 6-h data, yet removing them would eliminate about one third of the warm bias for the 3-h trajectories and one quarter of the bias for the 6-h trajectories. Redistributing the cold point locations within the colocated regions has an impact on the mean cold point temperature on the order of 0.01 K for both sets of lower resolution trajectories. Therefore, changes to trajectories' paths due to increased dispersion impact the cold point temperature by causing trajectories to cross the TTL outside of the cold trap regions, while the impact of trajectories skipping over the true cold point within these regions is negligible.

Excess dispersion also drives a warm bias by reducing trajectories' residence time in the TTL, thereby reducing the temporal temperature variability to which the trajectories are exposed. We quantify the trajectories' TTL residence time as the mean 340 K-to-400 K transit time, which decreases for DJF 2017 from 62 days for the 1-h trajectories to 56 days and 47 days for the 3- and 6-h trajectories, respectively. The mean cold point-to-400 K transit time also decreases from 37 days to 32 days and 26 days. Figure 3 shows the PDFs of the time at which each set of trajectories experiences their cold point, as well as the average of the minimum ERA5 temperatures at each latitude and height between 15°S and 15°N and 79 and 103 hPa (where 93% of trajectories experience their cold point). Qualitatively, increasing the residence time of the lower resolution trajectories would allow those that exit the TTL before the temperature drop centered around day 65 to experience that decrease in temperature, thereby correcting a portion of their mean warm bias. Starting the trajectories closer in time to the cold temperature anomaly could remove the warm bias for the 3- and 6-h trajectories, but this does not mean that the warm bias would be corrected for domain-filling trajectory studies. The mean warm bias would decrease by adding in periods when the lower resolution trajectories do not have a warm bias, but there will not be a period of cold bias for the 3- and 6-h trajectories that evens out the warm bias seen here.

We estimate the warm biases associated with the trajectories' decreased residence time by weighting the temperature (blue line) in Fig. 3 at each time step by the probability of there being a cold point at that time step for each set of trajectories (black, yellow, and red lines) and summing the results. That is, we calculate the dot product of the temperatures and the probability density functions in Fig. 3 in order to see how the faster transit through the cold point region creates warmer cold points. It is unreasonable to expect that all trajectories experience the absolute coldest point in the TTL during their transit, so this

calculation prevents overfitting to the three-dimensional temperature minimum by averaging over the layers where the cold point is likely to be observed. This approximation is justified by the increase in the probability densities in Fig. 3 at times when the zonal minimum temperature decreases, which indicates that trajectories are able to observe temporal temperature variability near the cold point. This calculation gives mean cold point temperatures of 185.3, 185.5, and 185.7 K for the 1-, 3-, and 6-h trajectories, respectively, so we estimate that the warm biases associated with the decreased residence time of lower temporal

resolution trajectories are 0.2 and 0.4 K. We stress that these values do not use Lagrangian cold points and are only estimates of how trajectories might be impacted by temperature fluctuations near the cold point, but they demonstrate the potential for a warm bias resulting from changes in residence time.

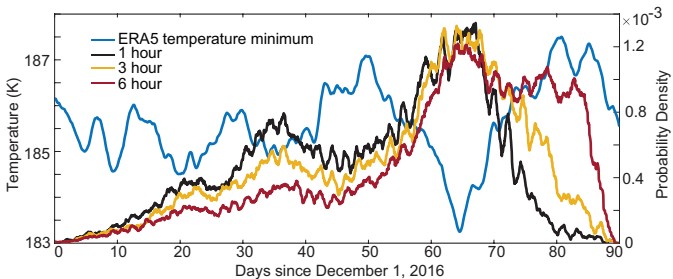

**Figure 3.** The daily running mean zonal minimum temperature averaged between 15°S and 15°N and 79 to 103 hPa for DJF 2017 from ERA5 data (blue), and the daily running mean PDFs of the time at which the cold point is observed by 1- (black), 3- (orange), and 6-h (red) trajectories. Trajectories are run in reverse from the end of this timeframe, so the distribution shifts to the left with increased transit times.

The remaining cold point temperature error is caused by the undersampling of the temperature field's temporal variability. We isolate this bias by subsampling the mean cold point temperature from the output of the 1-h trajectories every 3 and 6 hours.

Doing so removes the effects of the decreased transit times and shifted spatial distributions (i.e. removes the edge points) of the lower resolution trajectories by taking these directly from the 1-h trajectories. The cold point temperatures from the 3- and 6-h trajectories would be identical to this if the temporal resolution of the temperature was the only source of the warm bias, while differences between the two result from the decreased resolution of the wind field, as discussed above. The mean cold point temperatures from subsampling every third and sixth hour are 185.0 and 185.3 K, so 0.2 and 0.5 K of the 3- and 6-h

trajectories' warm bias can be explained by this output subsampling.

In summary, there are three drivers of the warm cold point temperature bias for trajectories with decreased temporal resolution: 1) the spatial extent of trajectories' sampling within the TTL (i.e. the inclusion of warm edge points), 2) the time at which trajectories encounter their cold point, and 3) the frequency of temperature sampling in time. As we state above, the warm bias

we calculate from the decreased residence time is only an estimate, plus it may be double counted in the warm bias of the

edge points. (If excess dispersion drives both of these, then it is possible that the trajectories at these points are warmer partly because they do not sample a full range of temporal temperature variability at their given location.) Therefore, these impacts cannot necessarily be summed linearly to reconstruct the total warm bias: for the 3-h trajectories, each of these effects have an estimated warm bias of about 0.2 K, which would sum up to greater than the total warm bias of 0.4 K, while the respective values of 0.3, 0.4, and 0.5 K for the 6-h trajectories sum up to the total warm bias of 1.2 K. These values are also specific

to DJF 2017, and their relative importance depends on the variability of the temperature and wind fields during the period of integration; regardless, we demonstrate that there is potential for each of these three effects to impact Lagrangian studies of the cold point.

### 3.1.2 Dispersion and transport across the tropopause

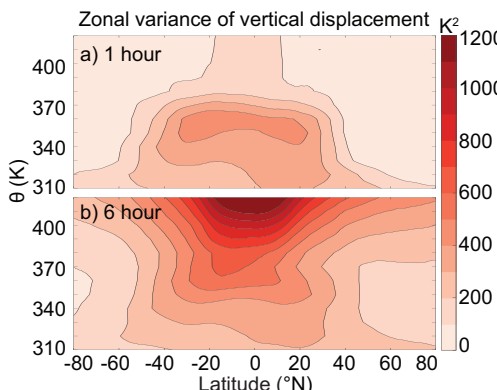

**Figure 4.** The zonal variance of potential temperature displacement for 1-h and 6-h trajectories run for 20 days with starting heights between 310 K and 420 K. Contours are at every 100 K$^2$. The zonal variance of 3-h trajectories is shown in Fig. S5.

Figure 4 shows the zonal variance of the displacement for 1- and 6-h trajectories initialized on interpolated isentropes

between 310 K and 420 K after 20 days of integration in January 2017. The variance below 350 K is similar between the two sets of trajectories, which suggests that decreasing the temporal resolution does not result in significant unrealistic vertical motion beneath the tropopause. Zonal variability in vertical motion is consistent with the three-dimensional structure of tropospheric circulation (i.e. the Walker circulation, mid-latitude storm tracks), so this variance is expected. The variance increases in the TTL for both 1- and 6-h data, and we argue that this should occur regardless of temporal resolution (though the extent to which

variance increases depends on temporal resolution). Tropospheric convection and lower stratospheric transport into the TTL are both zonally asymmetric and should therefore lead to a higher variance of vertical displacement for trajectories in this layer.

The variance above 380 K diverges for the two temporal resolutions shown in Fig. 4. For the 1-h data, the low variance in the tropics above 380 K reflects the zonal symmetry of air ascending in the upwelling branch of the BDC. For the 6-h data, the undersampling of the wind field causes lower stratospheric trajectories to be erroneously traced either across the TTL to the

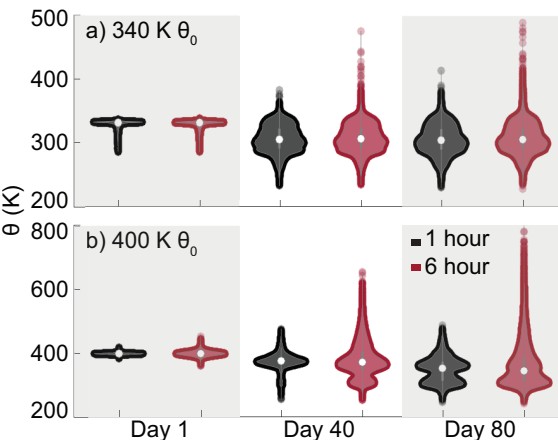

**Figure 5.** The distribution of trajectories' potential temperature for 1- (black) and 6-h (red) runs initialized on the interpolated 340 K and 400 K isentropes after 1, 40, and 80 days. Different y-axis are used to highlight the relevant ranges of potential temperatures.

troposphere or upwards into the stratosphere. We assume that the undersampling of the wind field driving the excess dispersion is random, so the increased variance should be symmetric and insensitive to the length of trajectory integration.

Figure 5 shows the distribution of the trajectory heights after 1, 40, and 80 days heights for trajectories initialized on the interpolated 340 K and 400 K isentropes with 1- and 6-h input data. This confirms the impact of dispersion above and below the tropopause: the distributions of the upper tropospheric (340 K) trajectories are nearly identical for the two temporal
resolutions, while the lower stratospheric trajectories are impacted by the choice of resolution. The outliers at the top of the 340 K distribution for the 6-h data after 40 and 80 days are driven by dispersion across the TTL, but the similarity of the distributions otherwise suggests that increased temporal resolution does not improve LAGRANTO's calculation of vertical transport in the troposphere or transport from the stratosphere to the troposphere in the deep tropics.

The impact of temporal resolution on vertical transport in the lower stratosphere and TTL is revealed by the differences
in the 1- and 6-h distributions for the 400 K trajectories. After 40 days, only 7% of 1-h trajectories have gone below 340 K, while 22% of 6-h trajectories have done so by this time. After 80 days, the fraction of trajectories traced to the troposphere is similar for the 6- and 1-h resolutions (47% and 43%), but their distributions throughout the stratosphere are different. The 1-h trajectories are confined to the lower stratosphere, while the 6-h trajectories have a long tail that extends up to 800 K, which reflects excess dispersion rather than a realistic representation of stratospheric circulation. These results together imply that the
excess dispersion resulting from insufficient temporal resolution causes lower stratospheric trajectories to be traced artificially from both above and below, thereby failing to resolve either the TTL transport barrier or the slow ascent of air in the lower stratosphere. The backwards Lagrangian trajectories studies discussed above completed integrations for at least 3 months, with some going for as long as 1 year (Fueglistaler et al., 2005; Liu et al., 2010). Increasing our integration length would not fix

the issues presented here because the trajectories that have been traced upwards into the stratosphere have been committed to a physically unrealistic path. Therefore, the fraction of lower stratospheric trajectories traced back to the troposphere may ultimately be too low for the lower resolution data despite being too high within the timeframe considered here.

As mentioned in Section 3.1.1, the average cold point-to-400 K transit time is reduced when the temporal resolution is reduced from 1- to 6-h as a result of dispersion in the TTL. The 6-h trajectories' cold point-to-400 K transit time is consistent with the 22.4 day transit time during DJF found by Fueglistaler et al. (2005) when using 6-h data, but that study notes that this transit time should be approximately 2 months based on age of air observations (Weinstock et al., 2001; Andrews et al., 1999). They conclude that their low transit time is due to the Brewer-Dobson circulation being too strong in their input data (ERA-40) and excess dispersion in trajectories that use analyzed wind fields in the lower stratosphere (van Noije et al., 2004; Schoeberl et al., 2003). ERA5 has a better representation of the BDC in the upper troposphere and lower stratosphere than its predecessors though (Diallo et al., 2021), so the similarity between our 6-h trajectory transit time to that of Fueglistaler et al. (2005) suggests that the issue of dispersion remains when using 6-h input data. We remind the reader that these results are based on kinematic vertical velocities, and that, although this effect is less important when using ERA5 data, diabatic velocities are potentially better suited for studies of the stratosphere (Schoeberl, 2004; Wohltmann and Rex, 2008; Liu et al., 2010; Li et al., 2020).

## 3.2 Vertical Resolution

The enhanced vertical resolution of the ERA5 dataset provides an opportunity to improve the accuracy of Langrangian trajectories by increasing the sampling of both the temperature and wind fields. As Fig. 1 shows, ERA5 has about 3 times as many vertical levels in the TTL as MERRA2, which is consistent with the ratio found by Tegtmeier et al. (2020). This enhancement is most relevant right around the cold point, which is highlighted by the blue contours in Fig. 1. The zonal mean temperatures are qualitatively similar between the two datasets otherwise, but LAGRANTO interpolates linearly in the vertical, so enhancing the vertical resolution in a region where the temperature profile is not linear should improve the trajectory temperatures. (The downsampled ERA5 data used here is shown in Fig. S1.) Enhanced vertical resolution should also decrease dispersion associated with undersampling of the wind field.

Figure 6 shows the cold point distributions for trajectories run during DJF 2017 with 1-h temporal resolution on the original ERA5 vertical grid with 137 levels and trajectories run on a reduced resolution vertical grid with 72 levels. The lower resolution trajectories have a mean cold point temperature of 185.8 K, or a warm bias of 1.0 K, and their cold point horizontal distribution is more diffuse: the fraction of cold points in the West Pacific cold trap decreases from 63.5% to 61.6%, and 92% of the lower resolution trajectory cold points are colocated with the full resolution cold points. The largest difference between the two sets of trajectories is the cold point pressure distribution in Fig. 6f. The fraction of trajectories that experience their cold points between 83 hPa and 87 hPa increases from 30.2% for the 137-level data to 50.4% for the 72-level data, which only has 1 level near the cold point at approximately 85 hPa.

As is discussed in Section 3.1, decreased sampling of the wind field can result in excess trajectory dispersion, which can alter the spatial and temporal distributions of the cold point. The edge points that result from this dispersion have a mean temperature of 188.0 K, while the colocated cold points have a mean temperature of 185.6 K. Therefore, removing the edge

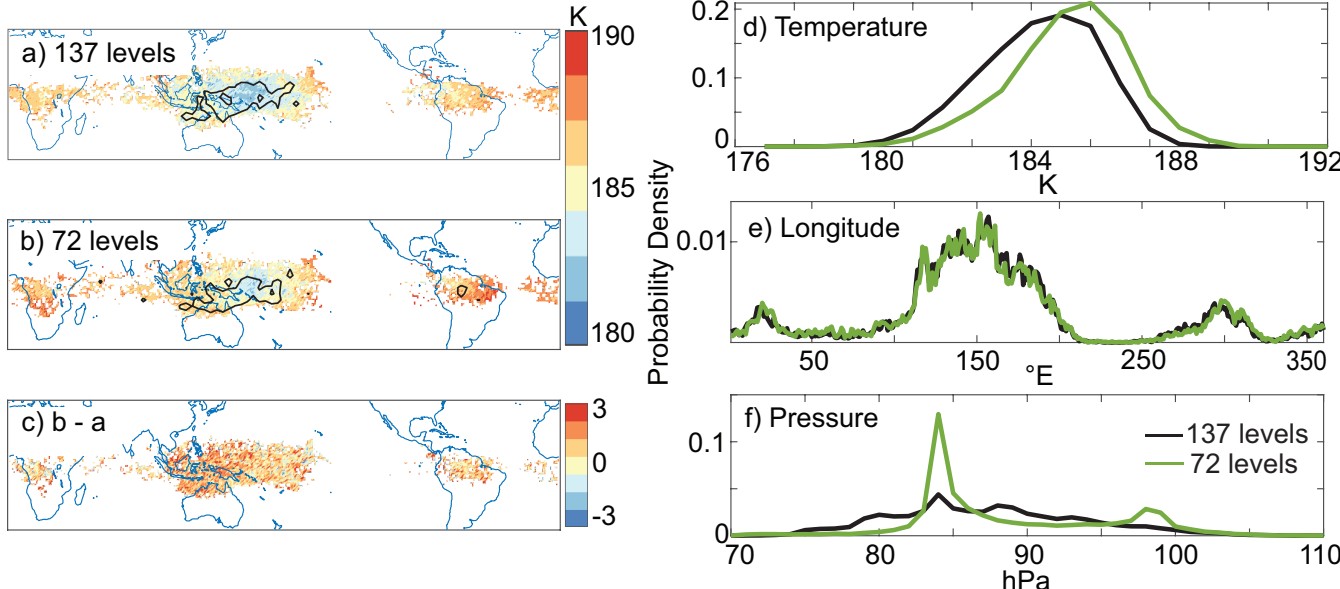

**Figure 6.** (a – c) Spatial distribution of the DJF 2017 trajectories' cold point temperature for 137- and 72- level data, and the difference in temperature between the 72- and 137- level trajectories. Black contours in a) and b) denote regions where the fraction of trajectories crossing grid cells is greater 0.03%. (e – g) Probability density functions of the cold point temperature, potential temperature, and pressure for 137 and 72 vertical level trajectories.

points would eliminate 0.2 K (about 20%) of the warm bias. As was the case with changes to the input data's temporal resolution, reweighting the cold points' spatial distribution within the colocated regions to match that of the full resolution
trajectories has a negligible impact on the mean cold point temperature. Quantifying the impact of dispersion on the temporal temperature variability sampled by these trajectories is complicated here by the different initial model setup required by the 72-level trajectories. These trajectories need an average of 12 days to reach 400 K from their 75 hPa starting height, and they are therefore able to experience the negative temperature excursion seen around day 65 in Fig. 3 despite having mean 340 K-to-400 K and cold point-to-400 K transit times of 48 and 25 days (14 and 12 days shorter than the full vertical resolution
transit times). Therefore, the warm bias that results from weighing the TTL temperatures by the time at which they experience the cold point is less than 0.1 K; in general, the decreased residence time of these trajectories should result in a larger warm bias by decreasing the likelihood that the trajectories are in the TTL during negative temperature anomalies.

    The remaining trajectory cold point warm bias can be explained by the undersampling of the temperature profile near the cold point. This is reflected by the PDFs of cold point pressure in Fig. 6f and the temperature profiles in Fig. 7: the 72-level
trajectories have a sharp peak in cold point pressure probability near 85 hPa, and the coldest temperature for these trajectories is centered around 85 hPa. The cold point temperatures measured by lower resolution trajectories are warmer throughout the layer because LAGRANTO's linear interpolation between the available temperatures is unable to produce the true cold point of the full vertical resolution data. As the dashed lines in Fig. 7 show, linear interpolation between the input data's vertical levels

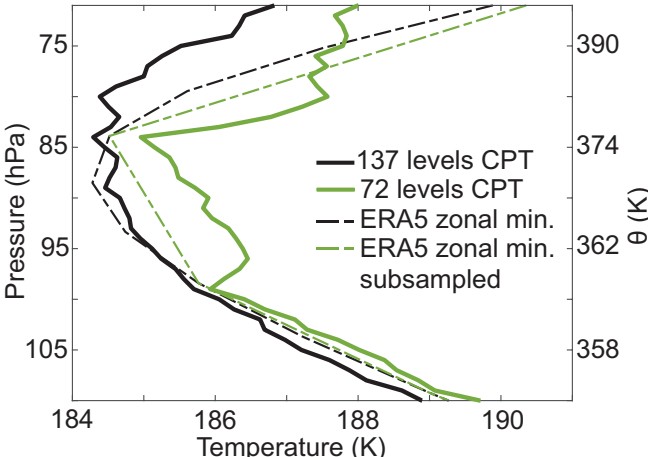

**Figure 7.** Temperature profiles of the DJF 2017 mean trajectory cold points measured by full (137-level) resolution and 72-level vertical resolution trajectories (solid lines), and the ERA5 time-average JF 2017 zonal minimum cold point temperature taken from the full grid (black dashed line) and the 72-level vertical grid (green dasehd line). The left axis is exact pressure from ERA5, and the right axis is the potential temperature calculated at those pressure levels with the DJF 2017 zonal mean temperature.

is bound to produce a warm bias for the lower resolution data. The average cold point temperature at 85 hPa is warmer for the lower resolution trajectories because the trajectories are not exposed to colder temperatures at other levels even when the 85 hPa temperature is relatively warm, so the value displayed in Fig. 7 averages over a greater range of temperature variability at that level.

The trajectory cold point temperature profiles in Fig. 7 incorporate cold point temperatures at all locations and times, so the lower resolution trajectories' profile includes the warming effects of edge points and residence time error. By removing these other biases from the mean cold point temperature error, we approximate that 0.65 K of warming results from the reduction in vertical temperature sampling alone. This is close to what would be expected from the bias of the input data alone: the average difference during JF 2017 (when 88% of trajectories experience their cold point) between the coldest temperature in the TTL from the full ERA5 data and the data subsampled at the 72 levels used by these trajectories is 0.47 K. Both of these bias estimates are greater than the improvement that Wang et al. (2015) found from increasing the vertical resolution of temperature in a Lagrangian trajectory model when using a wave resolving scheme (0.3 K) and when matching the input data with GPS temperature observations (0.2 K). That study used daily temperatures and 6-h winds, so its results may not have been as sensitive to changes in vertical resolution as our result is.

### 3.3 Horizontal resolution

As mentioned in Section 2.1, we found that our trajectory calculations were not impacted by improving the horizontal resolution of the input data from $1.0° \times 1.0°$ to $0.5° \times 0.5°$ or $0.25° \times 0.25°$ for trajectories run during DJF 2017. This can be seen in Fig.

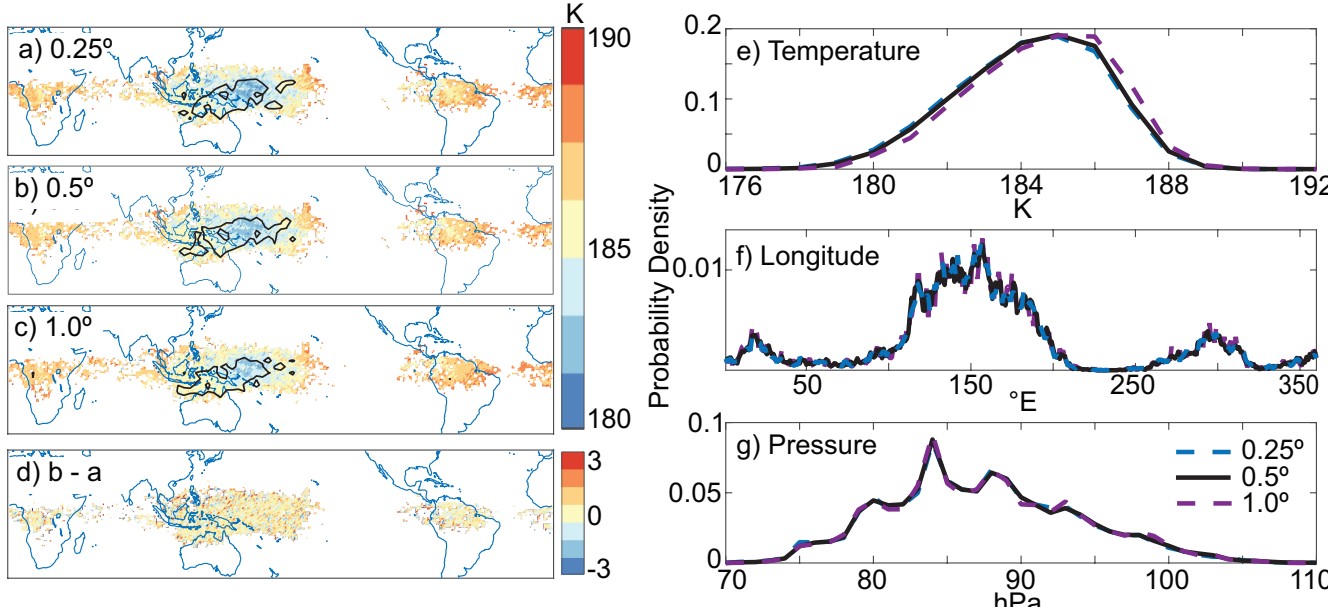

**Figure 8.** (a – d) Spatial distribution of the DJF 2017 trajectories' cold point temperature, and the difference in temperature between the 0.5° × 0.5° and 0.25° × 0.25° trajectories. Black contours in a), b) and c) denote regions where the percent of trajectories crossing grid cells is greater 0.03%. (e – g) Probability density functions of the cold point temperature, longitude, and pressure for 0.25° × 0.25°, 0.5° × 0.5°, and 1.0° × 1.0° horizontal resolution trajectories.

8, which shows the cold point temperature and horizontal distribution in the left column and PDFs of the cold point temperature, longitude, and pressure in the right column. We chose to display the difference between the 0.5° and 0.25° trajectories in Fig. 8d to emphasize the small difference between the two and justify our usage of 0.5° throughout this work. The spatial distribution of the cold points for 0.25°, 0.5°, and 1.0° trajectories shown in Fig. 8a–c are nearly identical: 95% of the cold points are colocated, and 64.1%, 63.5%, and 62.8% of trajectories experience their cold point within the West Pacific cold trap. The mean cold point temperatures are 184.7, 184.8, and 185.0 K for 0.25°, 0.5°, and 1.0°, respectively, and the mean cold point pressures are 87.3, 87.4, and 87.4 hPa.

Although the warm biases resulting from decreased resolution are small, we can decompose the warm biases into effects associated with undersampling of the wind and temperature fields. As discussed in Section 3.1, undersampling of the wind field can result in excess dispersion that alters trajectories' paths and reduces their residence time in the TTL. The mean 340 K-to-400 K transit times for 0.25°, 0.5°, and 1.0° trajectories are 62, 62, and 63 days, respectively, and the mean cold point-to-400 K transit time is 37 days for each set of trajectories. The estimated cold point temperature biases calculated from the time at which the trajectories encounter their cold point and the minimum TTL temperatures at those times are 0.01 and 0.02 K for the 0.5° and 1.0° trajectories. On the other hand, the altered horizontal distribution of cold points contributes about 0.1 K to the overall average cold point temperature for both sets of lower resolution trajectories by forcing the inclusion of edge points.

These points have a mean temperature of 186.9 K for both sets of lower resolution trajectories, while the colocated cold point temperatures are 184.7 and 184.9 K for the 0.5°, and 1.0° trajectories. Therefore, removing the edge points would eliminate almost all of the warm bias for the 0.5° trajectories and about a third of the warm bias for the 1.0° trajectories relative to the 0.25° trajectories.

The remaining warm bias comes from undersampling of the spatial temperature variability. We compare the tropical cold temperatures for different spatial resolutions in the following way: for each latitude between 15°S to 15°N and pressure level between 79 and 103 hPa, we find the minimum temperature (at any longitude) and calculate the mean over all latitudes and pressure levels. This is an Eulerian estimate of the difference in average cold point temperature due to the spatial variability. This increases by an average of 0.10 K and 0.35 K throughout DJF 2017 for the 0.5° and 1.0° data relative to the 0.25° data.

LAGRANTO uses a bilinear spatial interpolation in the horizontal to calculate trajectory positions between grid points though, which reduces the error introduced by undersampling of the horizontal temperature field. The decreased horizontal resolution of the temperature field nonetheless results in a warm bias in the mean trajectory cold point, but this error is small enough to be ignored by studies that need to optimize data storage or computation time.

### 3.4   Impacts on water vapor calculations

Studies that use the cold point of Lagrangian trajectories to reconstruct water vapor in the lower stratosphere (Fueglistaler et al., 2005; Schoeberl et al., 2013; Konopka et al., 2022) are subject to the biases associated with the spatial and temporal resolution of their input data described above. The moist bias is largest for the changes in temporal resolution discussed in Section 3.1: the saturation water vapor concentration at the cold point for DJF 2010 – 2019 increases from 1.53 to 1.66 and 1.94 ppmv when the temporal resolution of the input data decreases from 1- to 3- and 6-h, respectively. The decrease to the vertical resolution

discussed in Section 3.2 increases the cold point water vapor concentration by 0.24 ppmv, while decreasing the horizontal resolution from 0.25° to 0.5° and 1.0° as discussed in Section 3.3 increases the saturation water vapor concentration at the cold point of 0.03 and 0.10 ppmv, respectively.

        Due to the nonlinear relationship between temperature and water vapor mixing ratio, the shifted temperature distribution for the lower temporal resolution data results in a larger increase in water vapor than the increase calculated from the mean CPT

(positive water vapor anomalies from the warm tail of the CPT distribution are greater than negative water vapor anomalies from the cold tail). For example, the DJF 2017 1- and 6-h saturation water vapor concentrations are 1.51 and 1.82 ppmv when calculated with the mean cold point temperatures, while these concentrations are 1.59 and 1.95 ppmv when calculated with the full cold point temperature distributions. This means that the lower resolution trajectories' edge points have an outsized role in the moist bias: in DJF 2017 they comprise 11% of the 6-h cold points but drive 20% of the trajectories' moist bias.

Accurate reconstructions of water vapor also require the fraction of lower stratospheric air that has recently undergone dehydration at the cold point, which we cannot definitively comment on due to the insufficient length of our runs. We note, however, that the fraction is subject to the errors associated with the enhanced dispersion discussed in Section 3.1.2. This fraction will be too high for integrations using low resolution data for runs shorter than 90 days because of the enhanced dispersion, and it will likely be too low for longer runs due to the unrealistic portion of trajectories traced to the upper stratosphere (see upper

levels in Fig. 5b). Therefore, the portion of lower stratospheric air that has recently undergone dehydration at the cold point will be incorrect, and the reconstructed water vapor concentration will be biased accordingly.

The water vapor concentrations that would be obtained from assuming full dehydration at the cold point experienced by 1-h trajectories would be too low compared to observations, so other processes must be acting to hydrate the stratosphere. These could include microphysical (ice nucleation, particle growth, and aggregation) or advective (ice lofting, overshoot convection)

processes, or a combination of the two. For example, the timescale of ice particle formation and sedimentation may be longer than the length of time that air is exposed to the absolute minimum temperature along its path. In this case, the details of how fast ice particles form based on the availability of condensation nuclei and how fast they fall due to gravitational settling, which depends on their size distribution, will determine how much water is removed from the air and how much ice re-evaporates as the air encounters warmer temperatures and ascends. This has been explored previously (Schoeberl and Dessler, 2011;

Ueyama et al., 2014), but a detailed physical explanation is still lacking. The moist bias introduced by 6-h trajectories could approximate incomplete dehydration at the cold point, but this is a matter of chance and it cannot be generally assumed. Changes to the climate and atmospheric chemistry could alter the temperature and wind speeds at the cold point, as well the rate of ice nucleation and the ice particle size distribution. Therefore, even if 6-h trajectories can be used to reconstruct water vapor in recent observations, it is not guaranteed that this will be true in the future. A full picture of dehydration near the

cold point that includes these other processes needs to be considered in addition to the temperature history of air entering the stratosphere to determine how lower stratospheric water vapor will change in a changing climate.

## 4   Summary

Lagrangian trajectories' representation of the path of air through the TTL is degraded when the vertical and/or temporal resolution of the trajectory input data is decreased, but it is not significantly impacted by improvements to the horizontal resolution

of the input data beyond $1.0° \times 1.0°$, consistent with Bowman et al. (2013). This has implications for the temperatures sampled by the trajectories and ultimately for water vapor reconstructions in the lower stratosphere.

This impact is largest for changes to temporal resolution: lowering the instantaneous input data from 1- to 3- or 6-h resolution increases the mean cold point temperature for DJF 2010 – 2019 trajectories from 184.5 to 185.0 and 185.9 K. For the decrease in vertical resolution from 137 to 72 levels, the DJF 2017 mean cold point increases by 1.0 K, and the decrease in horizontal

resolution from 0.25° to 0.5° and 1.0° results in 0.1 and 0.3 K warm biases, respectively. These warm biases are caused by undersampling of both the temperature and wind fields. The majority of trajectories experience their cold points in the same cold trap regions regardless of resolution, but there is a warm bias within these regions for the lower resolution trajectories due to a shift in the time when the temperature field is sampled and the frequency of the temperature sampling in either time or space. The cold points erroneously measured outside of these regions by the lower resolution trajectories are also anomalously

warm: across all lower resolution trajectories, these edge points are an average 2.2 K warmer than those measured within the colocated regions.

The variance of displacement for trajectories initialized in the tropical lower stratosphere increases by an order of magnitude when the temporal resolution of the input data drops from 6- to 1-h, though the difference between 6- and 3-h is greater than the difference between 3- and 1-h, so future improvements could be small. This is consistent with Liu et al. (2010), which found kinematic trajectories with 6-h resolution to be overly dispersive relative to diabatic trajectories run with the same temporal resolution, and Li et al. (2020), which found that diabatic and kinematic trajectories run with 1-h data represented convection up to the tropopause layer better than those run with 6-h data (though we cannot comment on the performance of trajectories run with diabatic vertical velocities here). This excess dispersion degrades the trajectories' treatment of the tropopause transport barrier and decreases the mean trajectory residence time in the TTL from 62 days for 1-h trajectories to 56 and 47 days for 3- and 6-h trajectories. For runs lasting 90 days, the increased dispersion also increases the fraction of trajectories traced to the troposphere from the lower stratosphere. This trend will likely reverse at some point after 90 days though; the trajectories that are erroneously traced upwards into the stratosphere due to excess dispersion are unlikely to return to the lower stratosphere or cross into the troposphere on a timescale relevant to studies of troposphere–to–stratosphere transport.

The warm biases and excess dispersion of lower resolution trajectories will impact lower stratospheric water vapor reconstructions. The water vapor concentrations calculated based on saturation with respect to ice at the cold point using 3-h and 6-h trajectories are 0.13 and 0.41 ppmv higher than the water vapor concentration calculated with 1-h data. The 72-level vertical resolution trajectories have a moist bias of 0.24 ppmv relative to the 137-level trajectories, and the 0.5° and 1.0° trajectories have 0.03 and 0.10 ppmv moist biases relative to the 0.25° trajectories. Additionally, under- or overestimates of the portion of air recently traced to the troposphere due to excess dispersion will result in a composition that is biased by this error (i.e. too dry for too large of a fraction or too moist for too low of a fraction). Water vapor calculated from the highest resolution trajectories' cold points is too low relative to observations, which implies that other processes related to ice formation, sedimentation, and transport within the TTL need to be considered when projecting changes to lower stratospheric water vapor in a warming climate.

Here, we have shown that the statistics obtained from Lagrangian trajectories run across the TTL are sensitive to the temporal and spatial resolution of their input data. Although we cannot evaluate the trajectories calculated with 1-h data relative to the ground truth, it is clear from this work that lower stratospheric trajectories run with 3- and 6-h data are impacted by excess dispersion in the TTL, and their frequency of temperature sampling causes an additional warm bias. Similarly, input data with the full ERA5 137-level vertical grid smooths out the distribution of trajectories' cold point heights and cools their cold point temperature distribution, but it remains possible that additional vertical levels could further improve the trajectories' representation of troposphere–to–stratosphere transport through the TTL. Future studies of this region should consider these results when selecting input data, although work still needs to be done to compare trajectories run with 1-h kinematic vertical velocities to those run with diabatic vertical velocities. Of course, the highest temporal and spatial resolution would minimize error if storage and computing resources are not an issue, but reducing the horizontal resolution to $1.0° \times 1.0°$ will not meaningfully degrade results of Lagrangian trajectory studies within this region, and the temporal resolution can also be reduced to 3-h if a warm cold point temperature bias on the order of 0.5 K is acceptable.

*Code and data availability.* The ERA5 hourly data on native model levels from 2010 to 2019 used in this paper can be accessed through Copernicus Climate Change Service (C3S), https://apps.ecmwf.int/data-catalogues/era5/?class=ea. The source code for LAGRANTO powered by ERA5 data is available upon request from Michael Sprenger. The LAGRANTO trajectories and MATLAB code to analyze trajectories is archived and publicly available in Zenodo: https://doi.org/10.5281/zenodo.6410194 (Bourguet, 2022). The colormaps were created with the publicly available code for cbrewer (Charles, 2022), and violin plots made using publicly available code from Bechtold (2016).


*Author contributions.* SB: conceptualization, data curation, formal analysis, investigation, methodology, visualization, writing– original draft preparation, review and editing. ML: conceptualization, resources, supervision, writing– review and editing

*Competing interests.* The authors have no competing interests to declare.

*Acknowledgements.* Thanks to Bill Randel for helpful conversations, Michael Sprenger and Stephan Fueglistaler for help with calculating and interpreting trajectories, and Alison Ming and 2 anonymous reviewers for helpful feedback.

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
