# Peer review of "The Impact of Improved Spatial and Temporal Resolution of Reanalysis Data on Lagrangian Studies of the Tropical Tropopause Layer"

_Atmospheric Chemistry and Physics, 2022_

## Author Response (AR1)

We would like to thank the reviewers for their constructive and useful comments. We have addressed them as described below, and the paper has improved as a result.

**Reviewer 1**

It is unclear when you are using the DJF 2010 to 2019 integrations. Am I correct in thinking that most of the paper uses the 2017 integrations but the 2010 to 2019 integrations are in the supplementary material only? One suggestion would be to label (say A to G) the 7 experiments in Table 1 and use those labels in the figure captions.

The plots all use 2017 data, and the statistics that use are noted throughout the text 2010 to 2019. I have added a sentence at the beginning of Section 2.2 to clarify this:

"The figures presented in Section 3 contain data from DJF 2017, which is close to the average of all years; throughout the text we note when we are quoting either 2017 data or averages from 2010 – 2019. "

Page 2 L24 Worth explaining the water vapor tape recorder in a bit more detail as it will help a reader who is unfamilliar with this region of the atmosphere to understand the subsequent discussion.

Good point. I have added a bit to this discussion, and I hope that helps the reader.

"Subsequent work has confirmed the importance of the TTL in setting the humidity of the stratosphere by connecting  seasonal and interannual temperature fluctuations in the TTL to water vapor variability at heights throughout the lower and middle stratosphere with a time lag corresponding to the anomaly's transit time via the background circulation (the ``water vapor tape recorder," (Mote et al.,1996; Randel and Park, 2019). This correlation between local water vapor and the concentration at the tropopause some time earlier is strong up to about 30 km in the tropics is strong up to about 30 km in the tropics, at which point mixing and input of water vapor from the oxidation of methane dilutes the anomaly coming from the cold point."

Page 4 L107 ERA5 + ERA5.1

ERA5.1 data is only for 2000-2006, which we do not use here.

Page 6 L147 I am confused. I thought that you were running the trajectory model on hybrid pressure levels but here you are initializing the trajectories on isentropic surfaces. Are you interpolating from isentropic levels to initialise or is the model using isentropic levels?

The model initialized on interpolated isentropes and then runs on hybrid pressure levels. This new sentence in Section 2.1 should clarify this:

"The trajectories are run on pressure levels, so the trajectories' starting heights are interpolated onto the pressure levels corresponding to the 400 K isentrope in order to track how air parcels reach the 400 K isentrope."

Page 7 L167 Change 90 days to 3 months or vice versa for consistency

Thanks for catching this. It should say 90 days and is now fixed.

References: Please check your references carefully. There are various issues with the urls. E.g. Line 366 and many others

Thank you for catching this. I had copied each directly from a .bib file without checking.

**Reviewer 2**

This paper analyzes how the temporal and spatial resolution of model input data impact the cold point temperatures sampled by Lagrangian back trajectories. The trajectory simulations and comparisons done using ERA5 data are innovative and provide important information for the topics related to upper troposphere lower stratosphere transport. I suggest publication after the authors address my comments and make revisions accordingly.

> #1. The authors investigated the impact of the resolution of the input meteorological data on the cold point temperatures (CPT) sampled by trajectories. The conclusions are largely about how the distributions of CPT differ among different trajectory runs. This is useful information, however, it has its limitation since it does not show whether increased resolution results in an improved simulation compared to the real atmosphere. For example, which input resolution results in a more accurate CPT prediction and thus better water vapor prediction compared to the ERA5 water vapor itself or to the observation? The Lagrangian cold point temperatures are important for water vapor simulation, as stated by the authors, but the authors provided limited information (texts or figures) on water vapor predicted from the CPT. These are the major weaknesses of this paper. I highly recommend such analyses be added and discussed.

I have added a section to the results on this. This paper shows that a more accurate CPT lowers water vapor estimates. Therefore, the significance of this paper is its suggestion that there must be other processes that control dehydration in addition to the coldest temperature an air parcel encounters.

> #2. It is confusing that the authors claimed their trajectories as domain filling DJF trajectories, while they only released trajectories on five consecutive days in February. Technically, there are no trajectories initialized in January or in December.

I have removed the phrase "domain filling"

> Some specific questions related to this:

How long does it take for a parcel to reach the cold point level from 400 K? Do most parcels reach their cold point on nearby dates?

The average cold point-to-400 K transit times range from 25 days to 37 days. I have added transit time data for the trajectories throughout Section 3, including a figure (Fig. 3) of the PDF of the cold point time in Section 3.1.1.

The trajectories are initialized from the end of February. Given a climatological heating rate of 0.2-0.4 K/day in the TTL in boreal winter (Fueglistaler et al., 2009), it may take a parcel 50-100 days to reach 380 K. Therefore, the CPT the parcels sampled may occur in December or January. Since the authors only released parcels from the end of February, the cold point sampling may be limited to a few nearby days in January or December (This is only a rough estimate). If so, the CPTs are not technically DJF CPTs.

The trajectories have a February/January bias, and this bias varies based on input data resolution. This is an important point that I did not previously address. I have this sentence in Section 2.2 and address the relevant biases throughout Section 3:

"We do introduce a bias in the observed cold points towards the TTL conditions in late January and early February by initializing at the end of February, although trajectories still need to be run through December to trace them to the troposphere. As we will discuss in Section 3, this bias differs for each set of trajectories and needs to be considered when comparing trajectory cold points."

**3. Figures 2-4:**

Figure 2: There is no map or PDF for the 1-degree run in the left panel, though the texts mentioned it. Panel e shows a warm bias by the 1-deg run. The authors claim that the warm bias is due to horizontal temperature variability. However, panels a-c only explains the warm bias by the 0.5-deg run relative to the 0.25-deg run.

I have added maps and PDFs of the 1-deg run, but not the difference between 1-deg and the others. We add an explanation of the choice in Section 3.3:

"We chose to display data from the 0.5 and 0.25 trajectories here to emphasize the small difference between the two and thereby justify our usage of 0.5 for the remainder of this work"

For the right panel in Figures 2-4, I suggest adding PDFs of predicted water vapor and comparison to ERA5 water vapor or observation (as discussed in the first comment).

I have added a section on water vapor in the results. In this, I mention that this paper is not intended to reconstruct water vapor but instead raise concerns with the Lagrangian trajectory method for reconstructing water vapor. Because the higher temporal and spatial

resolution results in a colder cold point, and drier (too dry) stratosphere, simply calculating dehydration from the coarser resolution data is an oversimplification and likely to get plausible results for the wrong reason.

The ticks in y axis for panel d in Figures 2-4 are not consistent.

Fixed, thanks.

**4. How are the "collocated" CPT differences computed? This question can be applied to Figures 2-4.**

I have added an explanation of this in with Figure 2, and it applies to 3 and 4:

"For the distributions in Fig. 2a–c, the cold point locations are binned into 1° boxes and the average of all cold point temperatures within each box is taken; the difference of these averages for boxes where both 6-h and 1-h trajectories register at least 1 cold point (i.e. the cold points are colocated) is shown in Fig 2c."

1) Line 179: colocated-> collocated.

https://www.merriam-webster.com/dictionary/colocate
https://www.merriam-webster.com/dictionary/collocate
The original is correct.

2) Does collocated mean at the exact same grid point or just nearby grid points? Is there any spatial interpolation for nearby locations when computing the difference?

See above.

3) Are the CPT differences computed for points collocated horizontally only or three-dimensional? For example, Lines 229-230. There are clearly differences in vertical distributions judging from Figs. 4f-g.

This is a good point. The CPT differences are only done horizontally. We now clarify this and have added a figure to the supplemental (this is explained with Fig. 2 but is relevant to Figs. 3 and 4):

"Figs. 2a–c do not consider cold point height, although it generally decreases with increasing temperature and its changes follow the same spatial pattern as temperature (Fig. S9)"

4) What is the percentage of trajectories that collocate their CPTs for each run? How about the CPT locations from different runs that do not collocate? How much do those non-collocate points contribute to the temperature PDF shift (This seems to be addressed only for the temporal resolution experiment)?

I have added these numbers to Section 3. Thank you for the helpful suggestion.

5) For the horizontal resolution experiment, the authors ruled out the under sampling of the wind in causing the warm bias. Such a conclusion can be drawn if the authors confirm that they use CPT locations obtained from the 0.5 deg run and compare the temperatures from the 0.5 deg data and 0.25 deg data interpolated onto these CPT locations. Such a comparison directly tells whether the warm bias is purely caused by temperature variability in different datasets. But it is not clear how the difference in Fig. 2c is computed. Neither is there such calculation for the 1-deg run, which shows the largest warm bias. Is there a line for the 1-deg run in Fig. 2d? The lack of information makes the authors' explanations less convincing.

Thank you for bringing this up. Undersampling of both the wind and temperature fields create the warm bias. I have updated my results and discussion of horizontal resolution in Section 3.3.

6) Lines 186-192: "horizontal temperature variability" here the wording is a bit ambiguous and not consistent with previous texts. Do the authors mean temperature variability between different resolution data?

We have removed this phrase.

**Summary and final thoughts for comment # 4:** The method of the spatial-temperature difference analyses is not consistent or unclear throughout the paper, which makes their explanations of the temperature PDF shifts less convincing.

Hopefully our explanation of how comparisons are done has made this no longer a concern.

It seems from the paper, the CPT differences can be attributed to a) horizontal or vertical locational differences, b) given the same 3-D location, the temperature variability among different resolution data, c) differences in wind sampling. It would be helpful for the authors to include quantitative conclusions (for each of their experiment sets) on how much each factor contributes to the CPT bias in percentage and K.

Thank you for this idea. We now quantitatively discuss how each source of error contributes to the overall warm biases.

I also suggest consistency in the text when referring to temperature variability between different runs and other terms alike.

Thank you for the suggestion. I believe I have fixed this.

**5. Lines 290-291 vs Lines 295-296. The conclusions here stated seem to contradict each other. Lines 290-291: Do the authors mean far too large of a fraction of the lower resolution trajectories are traced to the stratosphere?**

Thanks for pointing this out. I have reworded this to make it more clear because actually these lines are not contradictory. Too large of a fraction of the lower resolution trajectories are traced to the upper troposphere, and the lower resolution trajectories are traced to the troposphere too quickly. So initially, the fraction is too large, but it is ultimately too small. I have reworded this for clarity:

"Increasing our integration length would not fix the issues presented here because the trajectories that have been traced upwards into the stratosphere have been committed to a physically unrealistic path. Therefore, the fraction of lower stratospheric trajectories traced back to the troposphere may ultimately be too low for the lower resolution data despite being too high within the timeframe considered here."

It would also be helpful to reference the figure number in these two paragraphs.

Thank you for the suggestion. I have added the figure number throughout the paragraphs.

Minor comments:

Line 126: The full title for CDO was not given.

Thank you for correcting this.

Lines 148-149: Is the vertical velocity w? Are the pressures and potential temperatures interpolated to the parcel's location or Lagrangianly integrated?

I have clarified this in Section 2.1:

"LAGRANTO computes parcel trajectories by integrating the velocity equation forward or backward through time using the three-dimensional kinematic wind field (vertical velocity is in units of Pa s-1). The wind vector is averaged between its value at the trajectory position and its value at the trajectory position at the following timestep three times before integrating the trajectory forward, and spatial interpolation is done using bilinear interpolation in the horizontal and linear interpolation in the vertical. The trajectories are calculated on pressure levels, and the potential temperature is recorded along the trajectories' paths."

Line 152: …relative humidity with respect to ice

Thank you for correcting this.

Lines 154-156: Are new trajectories released daily for 20 days in January or just released on one day in January? Which 20 days?

They are released on the last day of January and run backwards for 20 days. I've added an explanation of that at the beginning of Section 2.3:

"To explore the well documented vertical dispersion of kinematic trajectories, we ran a set of 20-day reverse trajectories on a global 0.5° x 0.5° horizontal grid initialized on pressure levels corresponding to isentropes every 10 K between 310 K and 420 K starting on the final day of January 2017 with 1-, 3-, and 6-h temporal resolution. "

Lines 310-316:

1) "warm bias … 0.5 K and 1.4 K".  vs Line 243: "…the mean cold point temperature for 1, 3, and 6 hour resolutions are 184.8, 185.2, and 186.2 K…" The numbers here do not match.

Thank you for the correction. This has been resolved.

2) "the shifted temperature distribution for the 6 hour data results in a 26% increase in water vapor… from the cold tail)." This estimate is not discussed in the main text before the summary.

I have added a section in the results on water vapor.

Lines 327-329: The fractions (0.58, 0.62, and 0.64) appeared in the summary but not in the main text. In section 2.3 the author stated that the trajectories for the dispersion experiment are released and tracked in January 2017. However, this paragraph is not consistent with those ("DJF 2010 to 2019"). Table 1 shows that the DJF 2010-2019 run is done for 1 hour resolution only.

I have removed the fractions from the summary and expanded the discussion of dispersion across the TTL in Section 3.3.2.

I have clarified that Table 1 only shows the runs for the cold point sensitivity experiments and added the 3 and 6 hour runs for DJF 2010-2019 to Table 1. The dispersion experiments described in section 2.3 are separate.

Lines 337-338: The authors actually can evaluate the water vapor predicted by the CPTs against ERA5 water vapor or observation.

These trajectories are not suitable for water vapor reconstructions, but they are suitable to identify errors that arise from various spatial and temporal resolutions. Please refer to the final paragraph of Section 3.4.

Comments on the summary section in general: The authors' summary of their results and the significance/impact of their findings is a bit repetitive and dispersive. For example, both the first and last paragraph of the summary section mentions that increasing horizontal resolution beyond 1 deg does not bring significant improvement. Another example is that the impact due to temporal resolution is mentioned sporadically and repetitively throughout the section.

References

Fueglistaler, S., Legras, B., Beljaars, A., Morcrette, J.-J., Simmons, A., Tompkins, A. M., & Uppala, S. (2009). The diabatic heat budget of the upper troposphere and lower/mid stratosphere in ECMWF reanalyses. Quarterly Journal of the Royal Meteorological Society, 135(638), 21–37. https://doi.org/10.1002/qj.361

**Reviewer 3**

This manuscript adopted the backward kinematic Lagrangian trajectory model to investigate the impact of spatial and temporal resolution of input meteorological fields on cold point temperature (CPT) simulations , therefore the impact on water vapor mixing ratio at entry into the stratosphere.  The quantitative evaluation results provide an important value on the simulations of troposphere to stratosphere transport and moisture distribution.  I suggest to consider publication after the authors appropriately address the comments.

General points:

1. The manuscript assessed the simulation of CPT with trajectory model and mentioned in the text the corresponding water vapor changes (e.g., 26% in summary part).  It is certainly worth showing the distribution of water vapor at cold point level and comparing with observations.

   The observations average over too large of a vertical domain to truly resolve the cold point (Bill Randel, personal communication), so comparing exact values between Clausius-Clapeyron dehydration and observations is not possible. The zonal mean water vapor concentration just about the cold point is meaningful here because of the horizontal motion and mixing that occurs within the TTL. We now discuss implications for water vapor in Section 3.4.

2. The setup for the LAGRANTO runs are not clear.
   1. In sec 2.2, the authors mentioned that the trajectories were calculated backwards for 3 months from the end of February to the beginning of December. Is this time at the release points or when the CPT was found along the trajectory?

   Trajectories start at the end of February and are tracked backwards until the first day of December, and the cold point can be encountered at any point during that time. I have clarified this in Section 2.2 (and added a discussion about biases related to when the cold point is observed later):

   "A set of trajectories was initialized at the final timestep of each day from February 24 to 28 and was integrated backwards to December 1 of the previous year

(integrations last between 86 and 90 days depending on start day). The trajectory cold point was taken at the coldest temperature recorded during the integration."

2. Lines 131-135, the release of parcels are not clear to me. "At the end of each day for 5 consecutive days". How many days in total per experiment from the end of February to the beginning of December? How to choose the 5 consecutive days?

The 5 consecutive days are the final 5 days of February (Feb 24, 25, 26, 27, 28). 5 sets of trajectories are run per experiment; 1 set is started at the end of each of these days. These days were chosen to maximize the length of the runs within the given months. I hope this clarification in Section 2.2 is helpful:

"A set of trajectories was initialized at the final timestep of each day from February 24 to 28 and was integrated backwards to December 1 of the previous year (integrations last between 86 and 90 days depending on start day)."

The figure S4 shows the PDF comparisons for two latitude ranges. How about the PDF of the cold point latitude?

There is very little temperature variability with latitude near the equator, so we did not include cold point latitude in our analysis. Qualitatively, you can see that the cold point latitude is not significantly impacted by the resolution changes explored in this paper.

3. lines 147-152, the trajectories tracked below 340 K were used for cold point analysis. What's the percentage of the trajectories used in this analysis compared to total initialized?

Thank you for suggesting that we address this point. I have added these numbers to Section 2.2:

"[W]e use trajectories that are traced below 340 K for our analysis of the Lagrangian cold point. This set of trajectories consists of 32–75% of the total initialized trajectories depending on the year and the resolution input data; interannual variability accounts for the majority of this variance, while input data resolution only drives a few percent change. For the DJF 2017 runs listed in Table 1, 46% of the 72-level vertical resolution trajectories, 53% of the 1-h trajectories run at each horizontal resolution, 58% of the 3-h trajectories, and 61% of the 6-h trajectories were traced below 340 K. (The 72-level trajectories have a smaller fraction due to their 75 hPa starting height.)"

For "the fraction of trajectories traced below 340 K at each timestep", is the black contours in figs 2-4 a&b related the fraction here? Or how was the fractions shown as black contours in figs 2-4 calculated?

> I have added an explanation of the black contours in Section 3.1, which applies to Figs. 2–4:

> "The contours shown in Figs. 2a–b denote the regions where the 1 boxes contain greater than 0.03% of the cold points observed by trajectories that are traced to the troposphere. (This is five times what the probability would be if the cold points were uniformly spaced across the tropics.)"

3. Figures 2-4
    1. In 2d, the PDF for 1.0 is missing; in 4d, the PDF for 3 hour is missing.

       > This had been done intentionally, but following your suggestion I've added the 1.0 and 3 hour PDFs in now.

    2. In 3, given the cold point pressure and potential temperature show a bimodal distribution, I suggest to check the same PDFs separately in three main regions: west Pacific Ocean, South America and central Africa to discuss the possible reason for the bimodal structure.

       > This was a good idea, but I have changed the model set-up such that there is no longer a bimodal distribution. If the reader is curious about the spatial distribution of cold point pressures, I have added a figure to the supplemental that shows maps corresponding to Figs. 2-4 (Fig. S09).

    3. Lines 216-219, is this related to the limitation of interpolation in lower vertical resolution grid?

       > This is related to the lack of data available for interpolation in the lower resolution grid. I have improved my explanation for this and included a figure of input data temperatures.

       > "The cold point temperatures measured by lower resolution trajectories are warmer throughout the layer because LAGRANTO's linear interpolation between the available temperatures is unable to produce the true cold point of the full vertical resolution data. As the dashed lines in Fig. 7 show, linear interpolation between the input data's vertical levels is bound to produce a warm bias for the lower resolution data."

Minor comments:

1. The saturation mixing ratio should add "relative to ice" .

   > Thank you for correcting this.

2. Line 221, it is 0.35 K in original reference.

   > Thank you for correcting this.

3. Consider add the calculated water vapor mixing ratio in text, beside the difference or percentage, such as lines 231 and 240.

Thank you for the suggestion; I have added a water vapor section to the results.

4. Lines 229-230 mentioned the difference first and later lines 241 gave the values. Consider put together or mention the mean values early.

Thank you for the suggestion; I have added the mean values earlier.

5. Line 248, are these values the calculated water vapor or the differences?

These are the calculated water vapor values. I have reworded this in the text (now in Section 3.4):

"the DJF 2017 1 and 6 hourly saturation water vapor concentrations are 1.51 and 1.82 ppmv when calculated with the mean cold point temperatures, while these concentrations are 1.59 and 1.95 ppmv when calculated with the full cold point temperature distributions"

6. Lines 249-255, is the horizontal coverage of the cold point sampled every 6 hours different from the 1 hour trajectories? Were the "edge" cold points still there?

The edge points are not included in this calculation. This allows us to isolate the bias introduced by the temperature sampling (the edge points are induced by undersampling the wind field). I have clarified this in Section 3.3.1:

"The remaining cold point temperature error is caused by the undersampling of the temperature field's temporal variability. We isolate this bias by subsampling the mean cold point temperature from the output of the 1-h trajectories every 3 and 6 hours. Doing so removes the effects of the decreased transit times and shifted spatial distributions (i.e. removes the edge points) of the lower resolution trajectories by taking these directly from the 1-h trajectories."

7. The summary part is not quite corresponding to the order of the results part. Such as summarized the variance in displacement first. Put discuss on temporal resolution ahead of vertical resolution and talk about temporal resolution again from line 327.

Thank you for the suggestion; I have reorganized this section accordingly.

---

## Author Response (AR2)

**Personal remark from Rolf Muller**

I think Fig. 7 is really helpful and it would be even more helpful if you managed to add a theta scale as an additional vertical axis (just as a suggestion)

Thank you for this suggestion. I have added a potential temperature axis.

**Report 1**

Publish as is.

Thank you for reviewing this paper!

**Report 2**

Lines 223-226 "The mean temperature... warm bias."
Lines 340-342 "The edge points ... cold point bias."
Lines 390-392 "These points have a... relative to the 0.25 trajectories."

It is not quite clear how the fractions of the warm biases are computed as described in the texts listed above. It is difficult to follow the authors' logic and calculations using the numbers given in the texts. I suggest revising the descriptions for clarity.
E.g. In the first analyses (Lines 223-226), how did the authors obtain the percentages 50% and 26%?
E.g., for the 6-h run, 11% trajectories have mean CPT of 187.9 K at the edge points, and 89% trajectories have mean CPT of 185.7 K in the collocated regions. This makes a weighted mean CPT of 185.94 K for the 6-h run. Compared to the 1-h run (184.8 K), this is a 1.14 K warm bias. The new CPT after removing edge points would be 185.7 K. Compared to the 1-h run (184.8 K), this is a 0.89 K warm bias. Assuming the rest of the bias is caused by the edge points, (1.14K -0.89K)/1.14K=21.93%. But this calculation fails to match the authors' 26%.

Our method for calculating the percent warm bias induced by edge points was (fraction of edge points)*(bias of edge points relative to higher res data)/(total bias). For this example, the calculation is 4640/43892*(187.9-184.8)/(186.0 - 184.8) = 0.27 (there was a small error in our code, so the 26% number was wrong by this method too). Your method is a direct comparison to what the average cold point temperature would be without edge points, which is easier to interpret, so we adapt it in the new manuscript. (The numbers differ slightly due to rounding though. We supply the mean CPT of 186.0, and a warm bias of 1.2 K, so the fraction of warm bias for the 6-h edge points should be (1.2 - 0.9)/1.2 = 0.25.) Thank you for this suggestion.

Line 353: Fig. 6g -> Fig. 6f

Thank you for catching that. This is now fixed.

**Report 3**

The authors have done a good job on addressing my previous comments and revising the manuscript. I recommend acceptance of this paper for publication.

Minor comment:
1. Check the Figs S3 and S4 in the supplemental files and the text.
Thank you for catching that; we had swapped the titles but forgot to change their labels. The labels are correct now. The references to these figures in the text did not need to be fixed.

2. Figure 6 misses the potential temperature.
I am not sure what this is referring to. The right column in Fig 6 shows the temperature, longitude, and pressure of the cold point, which is consistent with Figs 2 and 8. The first manuscript submission had potential temperature in the left columns of these figures, but we have removed that subplot in each figure to create space for the additional resolution comparisons in the left columns.

---

## Author Response (AR3)

**Main text:**

Line 213: I have added the sentence "Corresponding PDFs for these years are shown in Figs. S6–8."

**Supplemental figures:**

Figure S6: I have changed "Figure 4" to "Figure 2" in the description to reflect previous updates to the main text.

Figure S7: I have changed "Figure 4" to "Figure 2" in the description to reflect previous updates to the main text.

Figure S8: